# Creating Supported Plasma Membrane Bilayers Using Acoustic Pressure

**DOI:** 10.3390/membranes10020030

**Published:** 2020-02-18

**Authors:** Erdinc Sezgin, Dario Carugo, Ilya Levental, Eleanor Stride, Christian Eggeling

**Affiliations:** 1MRC Human Immunology Unit, Weatherall Institute of Molecular Medicine, University of Oxford, Oxford OX3 9DS, UK; 2Science for Life Laboratory, Department of Women’s and Children’s Health, Karolinska Institutet, 171 65 Stockholm, Sweden; 3Bioengineering Science Research Group, Faculty of Engineering and Physical Sciences, Institute for Life Sciences (IfLS), University of Southampton, SO17 1BJ Southampton, UK; D.Carugo@soton.ac.uk; 4McGovern Medical School, Department of Integrative Biology and Pharmacology, University of Texas Health Science Center at Houston, Houston, TX 77030, USA; ilya.levental@uth.tmc.edu; 5Institute of Biomedical Engineering, Department of Engineering Science, University of Oxford, Oxford OX3 7DQ, UK; eleanor.stride@eng.ox.ac.uk; 6Institute of Applied Optics and Biophysics, Friedrich-Schiller-University Jena, Max-Wien Platz 4, 07743 Jena, Germany; 7Leibniz Institute of Photonic Technology e.V., Albert-Einstein-Straße 9, 07745 Jena, Germany

**Keywords:** GPMVs, acoustic pressure, supported bilayers, plasma membrane vesicles, plasma membrane bilayers

## Abstract

Model membrane systems are essential tools for the study of biological processes in a simplified setting to reveal the underlying physicochemical principles. As cell-derived membrane systems, giant plasma membrane vesicles (GPMVs) constitute an intermediate model between live cells and fully artificial structures. Certain applications, however, require planar membrane surfaces. Here, we report a new approach for creating supported plasma membrane bilayers (SPMBs) by bursting cell-derived GPMVs using ultrasound within a microfluidic device. We show that the mobility of outer leaflet molecules is preserved in SPMBs, suggesting that they are accessible on the surface of the bilayers. Such model membrane systems are potentially useful in many applications requiring detailed characterization of plasma membrane dynamics.

## 1. Introduction

Artificial model membranes are useful tools for understanding cell membrane function and structure [1]. Their controllable composition facilitates the study of the role of specific molecules. Commonly employed model membrane systems include giant unilamellar vesicles (GUVs), hanging black lipid bilayers, and supported lipid bilayers (SLBs) [2]. In SLBs, unlike in free-standing membranes, a planar lipid bilayer of known composition is formed on a glass or mica surface [3]. For certain applications, SLBs have important advantages over free-standing bilayers. For example, they are much more straightforward to use for near field optical microscopy or atomic force microscopy [4], since they are immobile, present a large surface area and can be imaged for a long time [5]. Consequently, by incorporating receptor-like molecules on bilayers, SLBs can be used to study cell-cell interactions [5] in a very convenient format.

A disadvantage of the aforementioned model membrane systems is that they fail to mimic the compositional complexity of the cell membrane, being made up of only a few different lipid species. Consequently, giant plasma membrane vesicles (GPMVs) obtained from the plasma membrane of living cells have been used as an intermediate model membrane system between purely artificial membranes and the living cell [6,7]. This system has proven its usefulness in multiple fields ranging from membrane biophysics [8,9] to developmental biology [10] and drug delivery [11]. An important property of GPMVs is their capacity to exhibit lipid phase separation, allowing this phenomena to be studied in a more biologically relevant context [12,13,14,15,16,17,18]. In analogy to SLBs, there have also been efforts to create cell-derived (supported) planar model membrane systems. A common approach has so far been to disassemble cell-derived vesicles on glass surfaces by inducing the vesicle to burst with the help of additional artificial membrane patches [19,20,21], at air-water interfaces (thereby keeping the original cell membrane composition undisturbed) [22], or on polymer-supported lipid monolayers transferred from a Langmuir trough [23]. A common difficulty with all of these approaches is controlling which leaflet of the plasma membrane presents outwards (i.e., away from the support) which is essential when these systems are used to study molecular interactions.

To address this challenge, we propose a novel and straightforward approach for creating supported plasma membrane bilayers (SPMBs). It relies on the bursting of cell-derived GPMVs on a plasma-cleaned glass surface, using acoustic radiation forces that originate from an ultrasonic standing wave field [24]. Upon testing the diffusion of lipids as well as proteins in these SPMBs, we confirmed that the outer leaflet molecules are diffusive.

## 2. Materials and Methods

### 2.1. Cells, Lipids and Proteins

We used Chinese hamster ovary (CHO) cells (obtained from ATCC CCL-61) to create GPMVs. CHO cells were maintained in DMEM-F12 medium, supplemented with 10% FBS medium and 1% L-glutamine.

We purchased 23-(dipyrrometheneboron difluoride)-24-norcholesterol (Topfluor Cholesterol; TF-Chol) from Avanti Polar Lipids (Alabama, USA). Abberior Star Red-PEG-Cholesterol was obtained from Abberior. GPI-GFP is the plasmid published in ref [25]. Lypd6 was a gift from Dr Gunes Ozhan as in ref [10]. Cells were transfected with the plasmids using Lipofectamine 3000 as described in the manufacturer’s protocol.

### 2.2. Cleaning of Coverslips

To clean the coverslips, we performed oxygen plasma cleaning using a Diener Atto plasma system (Bielefeld, Germany). The coverslips were placed inside the plasma cleaner. After the air was pumped out, the oxygen was allowed to enter the chamber and plasma cleaning started. The coverslips were exposed to plasma for 15–30 s and transferred immediately into the acoustofluidic device for SPMB formation.

### 2.3. Preparation of GPMVs

The GPMVs were prepared as previously described [6]. Briefly, cells seeded out on a 60 mm petri dish (≈70% confluent) were washed with GPMV buffer (150 mM NaCl, 10 mM Hepes, 2 mM CaCl_2_, pH 7.4) twice. Then, 2 mL of GPMV buffer was added to the cells. After this, 25 mM PFA and 2 mM DTT (final concentrations) were added into the GPMV buffer. The cells were incubated for 2 h at 37 °C. Then, GPMVs were collected by pipetting out the supernatant. GPMVs were labelled by adding the lipid analogues to a final concentration of 50 ng/mL.

### 2.4. Acoustofluidic Device

The acoustofluidic device design and manufacturing protocols were taken from ref [26], with minor adaptations. The ultrasound (US) source consisted of a 13 mm × 30 mm × 1 mm piezoelectric element (PZ26, Meggitt PLC, Kvistgaard, Denmark), which was attached to a carrier layer using epoxy resin (RX771C/NC, Robnor Resins Ltd., Swindon, UK) cured at 30 °C for 24 h. The carrier layer consisted of a 0.8 mm thick machinable glass-ceramic material (Macor, Ceramic Substrates & Components Ltd., Newport, UK). A 0.2 mm × 12.0 mm (thickness × width) fluidic cavity was milled into the carrier layer, using a computerized numerical control (CNC) milling machine (VM10, Hurco Companies, Inc., Indianapolis, IN, USA). The lateral walls of this cavity were formed by a molded polydimethylsiloxane (PDMS) gasket (Sylgard^®^ 184, Dow Corning Corporation, Midland, MI, USA), which was manufactured by mixing a PDMS precursor and a curing agent (10:1 *w*/*w*), followed by degassing and curing at 90 °C for 1 h. The fluid cavity was sealed by a glass layer, consisting of a 75 mm × 25 mm × 0.15 mm glass coverslip (Logitech Ltd., Glasgow, UK). A micro-milled Perspex^®^ manifold was manufactured for integrating the device with inlet/outlet tubing, to deliver the suspension of GPMVs within the fluid cavity. A metal frame with a central cut-out was employed to achieve stable contact between the layers, whilst providing optical access for confocal microscope imaging. A one-dimensional (1-D) transfer impedance model implemented in MATLAB^®^ (The MathWorks Inc., Natick, MA, USA) was employed to design the thickness of each layer, in order to achieve the desired properties of the acoustic pressure field within the fluid cavity [27]. Since the thickness of the glass layer (0.17 mm) was significantly lower than the US wavelength (~1.9 mm), a significant proportion of the incoming acoustic energy was reflected at the glass-air boundary, resulting in a minimum in the acoustic pressure at this location (Figure 1B). Thus, the resulting primary acoustic radiation force acting on the GMPVs within the fluid cavity was directed towards the glass surface. The device was operated at its first thickness resonance of ~0.75 MHz, at a driving voltage of 25 V peak-to-peak. A frequency sweep of ± 0.025 MHz (sweep period: 50 ms) centred on the resonance frequency was applied, to ensure stable operation of the device over time. Upon loading the device with GPMVs, the piezoelectric element was actuated by a radio frequency (RF) power amplifier (55 dB, Electronics & Innovation, Ltd., Rochester, NY, USA), driven by a sine-wave from a signal generator (33220A, Agilent Technologies Inc., Santa Clara, CA, USA). An oscilloscope (HM2005, Hameg Instruments GmbH, Mainhausen, Germany) was used to monitor the applied voltage and US frequency. A continuous ultrasound wave was applied until the GPMVs’ bursting was observed, which took approximately 10–20 s.

### 2.5. Imaging and FCS

The SPMBs were imaged in GPMV buffer. All imaging was done at room temperature (21–23 °C) and on plasma-cleaned glass slides with a thickness of 0.17 mm. The samples were imaged with a Zeiss LSM 780 (or 880) confocal microscope. Topfluor, DiO and GFP were excited with 488 nm and the emission was collected with 505–550 nm. Abberior Star Red was excited with 633 nm and the emission was collected with 650–700 nm.

FCS on SPMBs was performed using a Zeiss LSM 780 (or 880) microscope (Jena, Germany) and a 40× water immersion objective (numerical aperture 1.2), as described previously [25]. Briefly, before the measurement, the shape and the size of the focal spot was calibrated using Alexa 488 and Alexa 647 dyes in water in an 8-well glass bottom (#1.5; 0.17 mm) chamber. To measure the diffusion on the membrane, the laser spot was focused on the SPMBs by maximising the fluorescence intensity. Then, 3–5 curves were obtained for each spot (five seconds each). The obtained curves were fit using the freely available custom-built FoCuS-point software [28], using the 2D and triplet models.

## 3. Results

Figure 1 shows a schematic of the procedure of creating supported plasma membrane bilayers (SPMBs). Giant plasma membrane vesicles (GPMVs) were placed on plasma-cleaned coverslips and their bursting induced (Figure 1A) by accelerating them towards the glass surface using acoustic radiation forces, within a custom-built acoustofluidic device. This device is based on a ‘thin-reflector’ resonator configuration, as described in references [24,29]. It comprises a 1 mm thick ultrasound generator (transducer) coupled with a 0.8 mm thick ceramic carrier, a 0.2 mm thick fluid layer containing a suspension of GPMVs, and a 0.17 mm thick cover glass (Figure 1B). The device is operated at the first thickness resonance of its layered structure (at a frequency of 0.75 MHz), resulting in acoustic pressure minima positioned at the solid-air boundaries (Figure 1B,C) [24]. The acoustic pressure within the fluid layer gradually decreases towards the glass surfaces (Figure 1B,C), and the suspended GPMVs are thus subject to an axial acoustic primary radiation force.

This force drives the GPMVs towards the glass surface, where they eventually burst (Figure 2A,B). Fluid motion generated by spatial gradients in the acoustic energy field (known as ‘acoustic streaming’) may also exert an additional mechanical force on GPMVs and thus contribute to the bursting process. Figure 2A,B show the GPMVs before US is applied, and the SPMBs formed after US exposure, respectively. To test whether any unusual membrane topologies are present (such as partial formation of membrane stacks), we measured the total fluorescence intensity in GPMVs and SPMBs. Higher fluorescence intensities in the SPMBs would, for example, indicate multiple membrane layers or stacks. However, we observed very similar intensity values in GPMVs and SPMBs, as shown by the ratio of the fluorescence intensity in the SPMBs and GPMVs being close to one (Figure 2C). However, we also observed sporadic brighter spots in the SPMBs (Figure 2B,C). Z-stack imaging showed that these bright clusters are three-dimensional (3D) tubular structures (Figure 2D,E) that are localized above the membrane. This type of structure may be caused by the acoustic pressure field.

An important aspect is the topology of the resulting SPMBs, i.e., whether the upper accessible surface is the inner leaflet (inside-out bursting) or the outer leaflet (outside-out bursting) of the original GPMVs (see Figure 3A,B for both scenarios). This is crucial for molecular accessibility and diffusivity, and hence for studying receptor activities or cell-cell interactions. To assess this, we first performed fluorescence imaging experiments using three-dimensional confocal microscopy and GPMVs derived from Chinese hamster ovary (CHO) cells labelled with DiO. Besides complete bursting, we occasionally observed semi-burst vesicles (Figure 3C), which can only be possible when GPMVs are burst from the bottom and form a bilayer gradually instead of bursting from the top, suggesting an outside-out bursting scenario (i.e., the accessible leaflet of the SPMB is the outer leaflet of the GPMVs (Figure 3B)).

To further confirm this, we prepared GPMVs from CHO cells expressing cytoplasmic fluorescent green fluorescent proteins (GFP). After forming SPMBs from these vesicles, inside-out bursting (Figure 3A) may result in a loss of GFP fluorescence due to the escape of GFP. However, we observed residual GFP fluorescence underneath the SPMB (Figure 3D), indicating the capturing of cytosolic components by the bilayer, which is, again, possible in the case of outside-out bursting (Figure 3B).

Finally, we tested the diffusion of outer leaflet molecules as well as of lipids coexisting in both leaflets. In the case of the inside-out bursting scenario (Figure 3A), the outer leaflet molecules would be expected to be immobile as they would get stuck between the SPMB patch and the glass coverslip, while they would remain mobile in the outside-out scenario (Figure 3B). We specifically measured the mobility of the following fluorescently tagged molecules in the SPMBs using fluorescence correlation spectroscopy (FCS): (i) Abberior Star Red-labelled cholesterol with a PEG linker as an outer leaflet lipid [30,31], (ii, iii) GFP-labelled glycosylphosphatidylinositol (GPI, ii) [32] and Lypd6 (a full-length GPI-anchored protein [10], iii) as two outer leaflet membrane proteins, (iv) Topfluor-labelled cholesterol as a lipid analog that partitions in both leaflets [31]. All of these components appeared to be mobile in the SPMBs (Figure 3E), confirming the outer leaflet molecules are accessible on the surface and not stuck between the bilayer and the glass slide.

## 4. Conclusions

We herein present a novel way of creating supported plasma membrane bilayers (SPMBs), employing acoustic radiation pressure to push GPMVs towards plasma-cleaned coverslips, to induce bursting. The control experiments showed a likely bursting mechanism that keeps the outer leaflet molecules accessible and mobile. All outer leaflet probes that we tested were still diffusing freely in the SPMBs. This is a very useful property, because such a system can present surface molecules and thus be used as a platform to study receptor-ligand interactions and signalling. Importantly, this methodology does not rely on extra synthetic lipid elements to trigger bursting, thus it preserves the native composition of the cell membrane. However, it is important to note that this system does not provide full asymmetry of the native cellular membranes. For instance, the lipid asymmetry is already partially broken in GPMVs [33] and further lipid flipping during SPMB formation might occur, and is yet to be studied. Nevertheless, the proteins are likely to keep their asymmetric distribution as well as mobility. The diffusivity of the molecules that we tested are comparable to those in the native cell membrane. Therefore, we believe that this system will be useful for several applications in cell and membrane biology.

In the future, many other features of SPMBs need to be characterized, such as membrane surface topology (e.g., multilamellar patches), stiffness or tension (e.g., using atomic force microscopy) or its molecular content (e.g., using lipidomics).

## Figures and Tables

**Figure 1 membranes-10-00030-f001:**
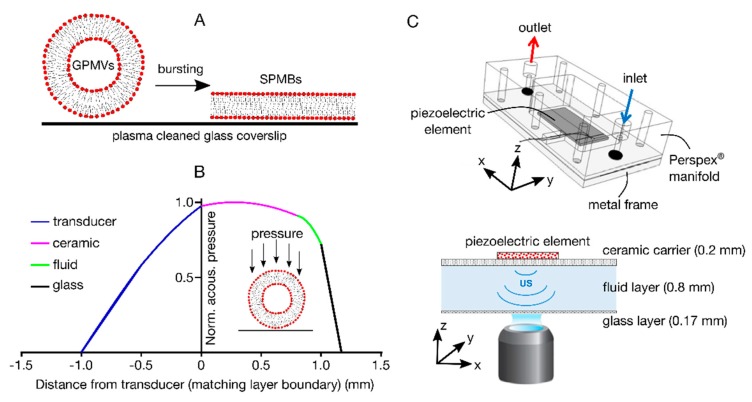
Overview of the experimental procedure. (**A**) Schematic of SPMB production from GPMVs. (**B**) Normalised acoustic pressure across the different layers of the device (at an operating frequency of 0.75 MHz), obtained from 1-D transfer impedance modelling. The device consisted of a 1 mm thick ultrasound generator (transducer, blue), a 0.8 mm thick ceramic carrier (red), a 0.2 mm thick chamber for the GPMV-containing fluid (green), and a 0.17 mm thick plasma cleaned microscope cover glass (black). Operating the device at its first thickness resonance generated a primary acoustic radiation force onto the GPMVs within the fluid, which was directed towards the glass surface (magenta area). (**C**) Top: Schematic representation of the acoustofluidic device assembly (isometric view), showing the Perspex^®^ manifold containing inlet/outlet ports, the metal frame, and the piezoelectric element generating the ultrasound (US) field. Bottom: Cross-sectional view of the piezoelectric element (US source), ceramic carrier layer, fluid layer, and glass coverslip. The thickness for each layer (in mm) is reported in brackets. These layers are kept in close contact with each other by screwing together the Perspex^®^ manifold and the metal frame.

**Figure 2 membranes-10-00030-f002:**
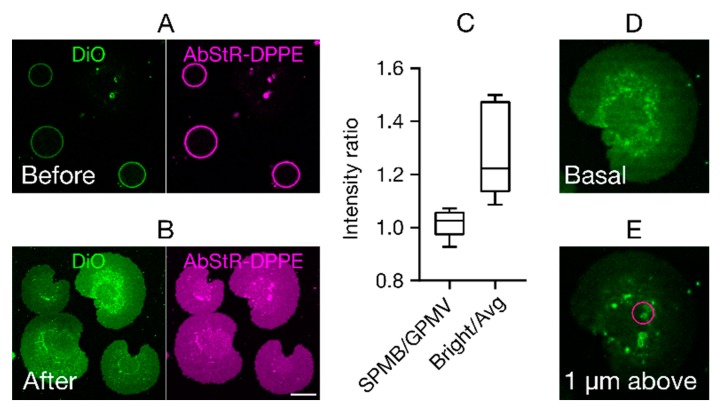
Supported plasma membrane bilayers (SPMBs). (**A**,**B**) Representative confocal microscopy images of the equatorial plane (**A**) of the GPMVs before and (**B**) of the SPMBs on the cover glass after applying the acoustic pressure field. Membrane labelling was done by the membrane dye DiO (green) and the fluorescently labelled fluorescent lipid analog Abberior-Star Red (AbStR; magenta) DPPE. Scale bar is 10 µm. (**C**) Fluorescence intensity ratios. Intensities in SPMBs divided by those in GPMVs (SPMB/GPMV) and in bright spots in SPMBs divided by average intensity outside those spots (Bright/Avg); “1” indicates equal intensities. (**D**,**E**) Representative confocal images of SPMBs (**D**) at the basal plane and (**E**) 1 µm above the basal plane, showing bright tubular structures above the membrane plane, causing the appearance of bright spots.

**Figure 3 membranes-10-00030-f003:**
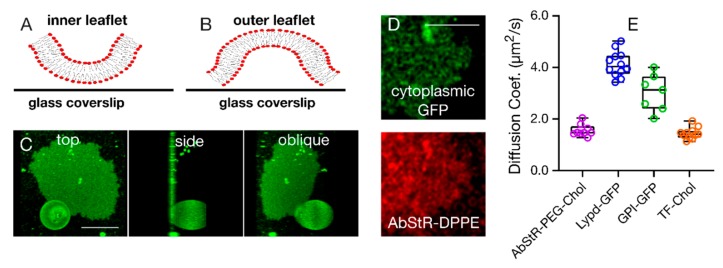
Bursting topology of SPMBs upon application of the acoustic field. (**A**,**B**) Inside-out (A, inner leaflet is facing upwards and is thus accessible) and outside-out bursting (**B**) (outer leaflet is facing upwards and is thus accessible) scenarios. (**C**) Representative confocal image of a SPMB with a semi-burst GPMV (right: top x-y view, middle: side y-z view, left: oblique 3D-rendering). (**D**) Confocal microscopy top view of a representative SPMB membrane-labelled with a fluorescent lipid analog (red, Abberior Star Red DPPE) and of cytoplasmic GFP expressed in cells and thus inside the GPMVs, indicating that the cytoplasmic GFP got captured underneath the bilayer upon bursting (i.e., outside-out bursting scenario). (**E**) Diffusion coefficients of various outer leaflet molecules as determined by FCS on the SPMBs. Scale bars are 10 µm.

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
