# Peer review of "Creating Supported Plasma Membrane Bilayers Using Acoustic Pressure"

_membranes, 2020, doi:10.3390/membranes10020030_

Round 1
Reviewer 1 Report
In this paper, the authors report a new approach for creating supported plasma bilayers.
This report is of good quality and well-written, allowing the reader to follow easily the protocol and the experiments reported by the authors.
While I have no issue with this draft and I still have a couple of quick remarks that the authors should address prior the publication:
- The authors mention on several occasions that the GPMVs burst on a plasma-cleaned glass surface, and it seems to be relatively important to have a pristine surface. Yet, no description of the cleaning procedure is given (what plasma cleaner was used, for instance?).
- In addition to the previous remark, Fig. 1 suggests that H2SO4/H2O2 solution (aka piranha solution) was also used conjointly with plasma to clean the glass surface. Except for this figure, no mention of the piranha solution is made elsewhere in the manuscript. As already mentioned, I think the authors should add a description of the protocol they used to clean the glass surface.
- In section 3.3, the authors state that "a significant proportion of the incoming acoustic energy is reflected at the glass‐air boundary, resulting in a minimum in the acoustic pressure at this location (Fig. 2C)" (lines 161-162). While this assertion makes sense, I can hardly see how it related to the actual figure 2C, which shows a semi-burst GMPV. If the authors refer to fig. 2C, they should rephrase their statement, so it is easier to see the relation between the two.
Reviewer 2 Report
The manuscript entitled „Creating supported plasma membrane bilayers using acoustic pressure” reports on the generation of supported lipid layers out of giant plasma membrane vesicles (GPMVs) obtained from the plasma membrane of living CHO cells. The application of acoustic radiation forces in a custom-built fluidic device triggers bursting of the GPMVs. The authors claim that this method constitutes a new method for the generation of supported lipid membranes comprising of a natural plasma membrane composition. Moreover, the study, performed by imaging (confocal microscope) and FCS measurements, suggests a directed outside-out orientation of the supported membrane with respect with the lipid leaflet orientation of the GPMV.
However, the authors have to clarify several issues and provide better explanations before the paper will be ready for publication.
1) The authors claim to: “present a novel way of creating supported plasma membrane bilayers (SPMBs) employing acoustic radiation pressure to push GPMVs towards plasma‐cleaned coverslips for bursting.” Unfortunately, it will hardly be possible to repeat these measurements because the information given in section 3.3 and Ref. 30 is not sufficient. A systematic drawing of the custom-built fluidic device would be very helpful.
2) Ultrasound (US) is a very commonly used method for liposome fabrication. US application causes the formation of unilamellar vesicles out of multilamellar ones by breaking-up and subsequent fusion. Moreover, for proteoliposome generation US waves promote the opening of liposomes and hence, membrane proteins insert into the lipid bilayer. In all these processes (breaking vesicles with subsequent fusion) it cannot excluded that lipids from the inner leaflet are after subsequent fusion arranged in the outer leaflet and vice versa. How can the authors be sure that there is no intermixing of “inner” and “outer” lipids during continuous US wave application for 20 seconds?
3) Fig. 1A and Fig. 2A, B schematically depicted the SPMBs formed on the glass coverslip. These schemata differ between each other with respect to the planarity of the formed SPMB. The latter is important for explaining the orientation (outside-out versus inside-out). A complementary method like, e.g., atomic force microscopy (AFM) would help to clarify this issue.
4) Fig. 1C shows images of the equatorial plane of the GPMVs before and of the SPMBs on the cover glass after applying the acoustic pressure field. Why is the fluorescence intensity for the bilayer structure (SPMB, thickness 7 to 10 nm) much higher than the fluorescence intensity of the GPMV in the focus of the microscope? Why show the SPMBs such a high inhomogeneity in terms of fluorescence intensity? How can one explain this high degree of inhomogeneity? Finally, the question raises: Is this actually a bilayer or more likely does it comprise of stacks of bilayers?
5) The authors investigated the topology of the resulting SPMBs (see Fig. 2). The authors claim that the GPMVs burst only from the bottom. However, Fig. 2C might also show a membrane patch with an adsorbed GPMV. Indeed, at Fig. 2 C, side view, one can see the formation of edges on the top, which make burst from top conceivable. If both scenarios (burst from bottom and top) are happening at the same time, the images shown in Fig. 2 D constitutes not a proof for the exclusive outside-out arrangement. Cytosolic fluid could also accumulate underneath the inside-out oriented lipid patches. Most probably there are mixed patches of inside-out and outside-out topology present. The diffusion coefficient measurements, where TF-Chol (a marker for both leaflets) has approx. the same diffusion coefficient than AbStR-PEG-Chol, which is present only in the outer leaflet support this assumption. If there is only outside-out arrangement, where the inner leaflet is in close proximity to the glass surface, TF-Chol should show a significantly slowed down mobility (i.e., reduced diffusion coefficient).
6) There is no proof for the formation of lipid bilayers. No experimental data exclude the formation of stacks of bilayers. Measurements like, e.g., by AFM are necessary to clarify this issue.
7) Substantial improvement of the English of the text is necessary, in particular as the authors use passive voice very extensively.
8) The cited references are not in a uniform style. Upper and lower case at the journal titles are arbitrary used.
Round 2
Reviewer 2 Report
Comments 1 to 7 are sufficiently addressed. I particularly appreciate the additional figures, as they are very helpful to the readers.
Unfortunately, comment 8 has not addressed at all. No changes have been made. In contrast, one reference has been added, again in incorrect style. As the authors show no appreciation in this regard, I want to pass this issue over to the editors.
In section 3, the numbering of the subheadings is incorrect. Hence, text editing is necessary.
Author Response
Comments 1 to 7 are sufficiently addressed. I particularly appreciate the additional figures, as they are very helpful to the readers.
- We are glad that the Reviewer appreciates the changes.
Unfortunately, comment 8 has not addressed at all. No changes have been made. In contrast, one reference has been added, again in incorrect style. As the authors show no appreciation in this regard, I want to pass this issue over to the editors.
- We are sorry for having missed this point. We indeed corrected our EndNote reference library but the correct styles are not transferred to the manuscript. But now, we corrected all the references manually.
In section 3, the numbering of the subheadings is incorrect. Hence, text editing is necessary.
- We fixed this.